# Racial Disparities in the Heavy Metal Contamination of Urban Soil in the Southeastern United States

**DOI:** 10.3390/ijerph19031105

**Published:** 2022-01-19

**Authors:** Daleniece Higgins Jones, Xinhua Yu, Qian Guo, Xiaoli Duan, Chunrong Jia

**Affiliations:** 1Department of Public Health, University of Tennessee, Knoxville, TN 37996, USA; 2School of Public Health, University of Memphis, Memphis, TN 38152, USA; xyu2@memphis.edu; 3School of Energy and Environmental Engineering, University of Science and Technology Beijing, Beijing 100083, China; guo_guo0825@163.com (Q.G.); jasmine@ustb.edu.cn (X.D.)

**Keywords:** environmental racism, soil–metal contamination, environmental justice, environmental health

## Abstract

(1) Background: Field monitoring data for addressing the disproportional burden of exposure to soil contamination in communities of minority and low socioeconomic status (SES) are sparse. This study aims to examine the association between soil heavy metal levels, SES, and racial composition. (2) Methods: A total of 423 soil samples were collected in the urban areas of eight cities across six Southern states in the U.S., in 2015. Samples were analyzed using inductively coupled plasma–mass spectrometry (ICP–MS) for eight heavy metals. The association was examined with mixed models with the log-transformed metal concentrations as the dependent variables and rankings of low-income or minority percentages as the explanatory variables. (3) Results: Model results showed that soil metal concentrations were significantly associated with rankings of poverty and minority percentages. The cadmium concentration significantly increased by 4.7% (*p*-value < 0.01), for every 10 percentiles of increase in poverty rank. For every 10 percentiles of increase in minority rank, the soil concentrations were significantly up (*p*-values < 0.01) for arsenic (13.5%), cadmium (5.5%), and lead (10.6%). Minority rank had significant direct effects on both arsenic and lead. (4) Conclusions: The findings confirmed elevated heavy metal contamination in urban soil in low-income and/or predominantly minority communities.

## 1. Introduction

Growing evidence has documented disproportionate exposure to environmental pollution among low-income and minority populations [1,2,3,4]. Previous studies examined spatial distributions of environmentally hazardous facilities and revealed the clustering of these facilities in or near communities of color, and in poverty [5,6]. In the past years, numerous studies have found a significant association between environmental exposure and socioeconomic status (SES)/racial compositions [1,2,3,5]. These two types of studies mostly had ecological designs in which the unit of analyses was census tract [1,5,6,7], zip code [5,6,8], or county [5,6,9]. A few studies conducted environmental disparity analyses at an individual level [10,11,12], for the need to determine individuals’ health risks. Noticeably, most individual-level studies utilized body burdens of chemical toxicants measured by the U.S. CDC’s National Health and Nutrition Examination Surveys (NHANEs). The results showed inconsistent functions of SES on toxicant exposures [10,12].

Knowledge gaps remain in the large body of environmental justice (EJ) literature. Air pollution exposure dominates EJ research [1,3,5,6], despite the fact that humans are exposed to multiple pollutants via multiple pathways. This domination results from the easy availability of air pollution data measured by the national air monitoring network [13] and those modeled by the National Air Toxics Assessment (NATA) programs [14]. As a major exposure route, the ingestion of microbial and chemical contaminants in drinking water, food, diet, and soil receives little attention in EJ studies, possibly due to data scarcity. This gap impedes our understanding of disparate exposures to complex pollution mixture as a major contributing factor in the production of health inequities.

Heavy metals in soil represent a palpable environmental hazard to human health. Heavy metals are metallic elements with a density five or more times higher than water [15]. Some low-density metals and metalloids, e.g., barium, arsenic, and selenium, are often considered heavy metals given their toxicity. Heavy metals are known to lower energy levels and damage the functioning of the brain, lungs, kidney, liver, blood composition, and other important organs [16]. Long-term exposure also lead to the progression of physical, muscular, and neurological degenerative processes that imitate diseases, such as multiple sclerosis, Parkinson’s disease, Alzheimer’s disease, muscular dystrophy, and cancer [16,17]. Populations of minorities and those in poverty have higher risks of heavy metal-related diseases, such as less control over cognitive tasks, lower testing scores, and a decline in memory [18,19]. This health disparity may be attributable to higher exposure to soil heavy metals among disadvantaged populations [2,20,21].

Few EJ studies have evidenced disproportionate exposure to soil metals. Very few studies have measured soil metal concentrations in residential urban and rural locations with regard to specific contaminant sources [2,22,23,24]. Even more so limited is the soil metal data in low-income residential communities [21]. Only a few studies have researched disparity and soil–metal contamination [2,21,25,26,27]. The lack of soil contamination data hampers the environmental disparity analyses. As an alternative, previous studies only examined the geographic clusters of soil–metal contamination in proximity to point sources, such as industrial facilities [2,21,25].

Documentation of disparities in soil metal exposures would supplement the current air pollution dominant EJ research, help explain health disparities of metal-related diseases, and inform environmental remediations and integrations [25]. The objective of this study is to examine the association between soil heavy metal levels, SES, and racial composition across the Southeastern area of the United States (U.S.), as a part of the EPA’s Regional Urban Background Study. We hypothesize that neighborhoods with a greater percentage of minority and low-income bracket individuals will have a greater risk of exposure to heavy metal contamination. We also aim to identify the key built environmental factors that serve as mediators of this exposure-SES/race association.

## 2. Materials and Methods

### 2.1. Monitoring of Heavy Metals in the Surface Soil

This study utilized the soil heavy metal data from EPA’s Urban Background Study conducted in southern states from 2015 to 2016. The Urban Background Study was conducted to document background concentrations of metals in surface soils of urban areas in the Southeastern U.S. [28]. The eight cities under investigation were Memphis, TN, Chattanooga, TN, Columbia, SC, Gainsville, FL, Lexington, KY, Louisville, KY, Raleigh, NC, and Winston-Salem, NC. The soil sample collection adopted a systematic random sampling approach. In each city, a grid of 7 mile × 7 mile was applied in and around the urban center, and then divided into 0.5 mile × 0.5 mile cells, resulting in 196 cells. At least 50 cells were randomly selected, and a sample was collected within each selected grid using simple random sampling. The maps of the study areas and sampling sites of each city are displayed on the project’s website [28]. Each sampling location was carefully determined to be representative of the larger urban setting. The location should be in a public area and undisturbed. Broad-spectrum locations were considered, i.e., sampling locations consisted of both EJ areas and more affluent urban areas. A total of 423 soil samples were collected in the eight participating cities.

At each selected sampling location, a grab surface soil sample was collected from the upper 2 inches of the soil in the undisturbed soil horizon using a coring device. The soil was homogenized in a disposable aluminum pan and then sealed in a pre-labeled, certified clean sample container. The soil samples were delivered to EPA Region 4 Science and Ecosystem Support Division (SESD) Laboratory for metal analysis. Soil samples were digested with a nitric acid (HNO_3_) and hydrogen peroxide (H_2_O_2_) mixture, and then analyzed for seven metals: arsenic (As); barium (Ba); cadmium (Cd); chromium (Cr); lead (Pb); selenium (Se); and silver (Ag). This involved two steps: target metals were first screened using inductively coupled plasma–atomic emission spectroscopy (ICP–AES) following the SW-846 Method 6010C [29], and then confirmed and quantified using inductively coupled plasma-mass spectrometry (ICP-MS) following the EPA Method 200.8 [30]. The data was verified by EPA Region 4 SESD Analytical Support Branch (ASB) in accordance with the Laboratory Operations and Quality Assurance Manual (LOQAM) [31]. Details of the siting, sampling, and analytical methods were described in the Quality Assurance Project Plan (QAPP) of this study [32].

### 2.2. Socio-Demographic and Built Environmental Factors

The census block group (CBG) number of each sampling location was identified by overlaying the 2015 CBG map [33] on the sampling location maps in ArcGIS (Version 10.5, Esri Inc., Redlands, CA, USA). The CBG-level data about demographics, socioeconomic status, racial composition, and built environmental characteristics for each sampling location were then obtained from EPA’s 2016 EJScreen dataset [34]. The EJScreen database contained demographic and environmental health data from a multitude of publicly available sources, which enabled researchers to compare EJ in marginalized communities to state, regional, and national averages [35,36,37]. After reviewing previous studies [2,21,25,36], we selected the following variables for disparity analyses: percent of the minority (minority ranking), percent of households below the poverty line (poverty ranking), proximity to traffic, proximity to treatment storage and disposal facilities, proximity to major and direct discharges to water, proximity to national priorities sites, and proximity to risk management plan facilities. We used national percentile rankings of these variables from the EJScreen database to make fair comparisons among multiple cities and the results easier to interpret.

### 2.3. Statistical Analysis

The overall goal of the statistical analysis was to explore the association between soil heavy metal contamination and SES/race in this region. All the metals had levels above the detection limits in all the soil samples, and thus there was no need to treat non-detects. The distributions of metal concentrations were right-skewed; thus, they were natural log-transformed for the following analyses. For each metal, the disparity was first examined using a crude model in which the log concentration was the dependent variable and poverty rank or minority rank was the explanatory variable. The crude model would display the direct relationship between metal exposure and income/race. The resulting coefficient meant the change in log concentration per unit change in the explanatory variables. A coefficient >1 indicated positive association and vice versa. For the convenience of interpretation, we translated each coefficient into percent change in the concentration per 10-percentile change in the ranking of income/race. The models were performed in SAS (Version 9.4, SAS Institute Inc., Cary, NC, USA).

The built environmental variables might serve as mediators that explain the process through which metal exposure and SES/race are related. For example, low-income people often live near industrial facilities due to low housing prices [38,39], where the surrounding soil is more contaminated [40,41]. Thus, we used path analysis to examine the comparative strength of social and environmental factors on exposure to heavy metals [42]. Figure 1 displays the hypothesized model of the relationship among SES, race, environmental factors, and soil heavy metal contamination. The hypothesized model is a just-identified model, in which the number of free parameters exactly equals the number of known values, meaning zero degrees of freedom [43]. The path analysis was conducted using the ‘Lavaan’ package [44] in R version 3.5.3 [45]. Statistical significance was set at *p* < 0.01 for all the regression and path analyses.

## 3. Results

### 3.1. Descriptive Statistics for Minority and Poverty across the Southern U.S.

Poverty and minority showed varying patterns across cities in the southeastern region of the states (Table 1). The average percentile for poverty was highest in Memphis (86.87) followed by Columbia (71.17). The poverty percentile of 86.87 in Memphis tells us that the poverty percentage in Memphis was equal to or greater than 86.87% of the rest of the cities in the nation. The average poverty percentile was lowest in Lexington (46.91). The highest average percentile for minority was also in Memphis (87.62) followed by Louisville (78.91), with the smallest average percentile for minority in Gainesville (53.60). Memphis’ minority percentile reveals that the minority percentage in Memphis is equal to or greater than 87.62% of cities across the US.

### 3.2. Descriptive Statistics for Heavy Metal Levels in Soil

Heavy metals displayed varying abundances in soil in southeastern states (Table 2). The most abundant metals were lead (95.8 mg/kg) and barium (86.8 mg/kg). Chromium and arsenic showed medium abundances of 13.8 mg/kg and 5.14 mg/kg, respectively. Selenium, cadmium, and silver had low concentrations below 1 mg/kg. The coefficient of variation (COV) indicates the spatial variability of the metal concentrations. Arsenic displayed a large variability (COV = 291%), whereas chromium had limited variability (COV = 67%). Other metals displayed a substantial variability of 94–161%. As later analyses show, SES and racial factors were contributors to this spatial variability.

### 3.3. Direct Relationship between Soil–Metal Contamination and SES/Race

Soil metal concentrations displayed associations with poverty and minority (Table 3). The crude models showed soil metal concentrations had inconsistent relationships with poverty, but were mostly insignificant. Only the cadmium concentration significantly increased by 4.7% (*p*-values < 0.01), for every 10 percentiles of increase in poverty rank. For minority, crude associations were more stable and positive. Arsenic, cadmium, and lead soil contamination had a significant positive relationship with minority. For every 10 percentiles of increase in minority rank, the soil concentrations were significantly up (*p*-values < 0.01) for arsenic (13.5%), cadmium (5.5%), and lead (10.6%). The results from crude models suggest that soil–metal contamination did not differ across different SES levels except for cadmium, but was elevated in minority-concentrated areas. Further analyses by city, however, showed insignificant associations (Appendix A), possibly due to the small sample sizes.

### 3.4. Direct and Indirect Effects of SES and Race

We created seven trimmed models for each metal to evaluate the pathway between soil–metal contamination and SES, race, and built environmental factors. Models were trimmed by removing built environment variables if they did not have significant associations in the paths. A resulting example of the trimmed arsenic model can be seen in Figure 2, and the other models are displayed in Appendix A.

Proximity to sources mediated the relationship between SES/race and soil–metal contamination, as summarized in Appendix A. Proximity to risk management facilities and national priorities sites were the most common factors that mediated the relationship between SES/race and soil metal exposure. Thus, soil–metal contamination indirectly increases in poverty-filled and minority areas, mainly due to risk management facilities and national priority sites. Proximity to risk management facilities was a common factor for barium, cadmium, lead, selenium, and silver. This result hints that poverty-filled and minority communities have an increase in soil contamination of barium, cadmium, lead, selenium, and silver due to risk management facilities. There are slight differences in the most common mediators for the affects it has on specific heavy metals. Proximity to national priorities sites was a common factor that mediated the relationship between SES/race and barium, chromium, lead, selenium and silver. The next common mediator was proximity to treatment and disposal facilities, which was present for barium, cadmium, chromium, and lead. Hence, the soil–metal contamination of barium, cadmium, chromium, and lead indirectly increases in poverty-filled areas and minority areas due to proximity to treatment and disposal facilities. Proximity to traffic was a common factor that mediated the relationship between SES/race and soil contamination for arsenic, barium, and silver. The least common factor was direct water discharge, only present for arsenic and barium. The results from direct water discharge indicated that this mediator was least important in indirectly assessing increasing soil–metal contamination in minority and poverty-filled areas. Despite the mediating effects of the built environment factors, SES and race still showed direct effects on soil–metal contamination. The Minority factor displayed significant direct effects for arsenic, chromium, and lead, whereas poverty showed direct effects for chromium, selenium, and silver.

The standardized estimates for direct, indirect, and total effects between minority/poverty and soil–metal contamination are presented in Table 3. Poverty rank had negative direct effects on soil–metal concentrations, i.e., higher poverty rank was associated with lower metal concentrations, and the association was significant for arsenic (β = −0.14) and selenium (β = −0.20) (*p*-values < 0.01). In terms of indirect effects, poverty rank showed significant indirect effects for cadmium (β = 0.15), selenium (β = 0.23), and silver (β = 0.18). Poverty did not display any significant total effects. These results were slightly different from our crude analysis, where poverty held only a significant positive direct effect on cadmium. Here, we found that the significant effect for cadmium was due to built environment characteristics, which was the indirect effect. Interestingly, we saw that selenium showed a significant negative direct effect, but a positive indirect effect through the mediators, or built environment characteristics.

Minority rank had significant direct effects on arsenic (β = 0.14) and lead (β = 0.15) (*p*-values < 0.01). The positive coefficients indicate that communities with higher minority percentages had higher soil–metal contamination. Minority also exhibited significant indirect effects for arsenic (β = 0.18), barium (β = 0.12), and lead (β = 0.09). Combining direct and indirect effects, minority showed significant total effects for arsenic (β = 0.32), barium (β = 0.18), and lead (β = 0.24). An interesting fact is that barium is now significant in terms of total effects, even though we did not see a significant direct effect. This new result is also due to the indirect effect of the built environment characteristics.

## 4. Discussion

Our analyses suggest that areas with a higher population of minorities have an increased exposure to heavy metal contamination in their environments. Though, on a lesser scale, there is also significant risk of select heavy metals for poverty-stricken areas. Path analyses also revealed that these relationships were mediated by proximity to emissions sources, confirming the pathways of SES/race -> proximity to emission sources -> soil–metal contamination.

### 4.1. Emission Sources and Soil Contamination

Proximity to emission sources should be a major cause of soil contamination, as confirmed by our results. The positive associations, indicated as indirect effects in Table 4, suggest higher soil–metal contamination near emission sources, which has been reported by numerous studies [46,47,48,49,50]. Specifically, traffic proximity, treatment and disposal facility proximity, and proximity to risk management facilities presented significant positive relationships for all metals (Appendix A). Vehicular emissions, such as tire degradation and brake wear are sources of metals such as barium and chromium [51,52]. Opposite relationships, however, were found for proximity to direct water discharge and national priority sites. A nearby water resource can decrease the soil absorption of heavy metals [53], confirming the negative association found for proximity to direct water discharge. In addition, national priority sites are prioritized for soil, water, and air contaminated by organic chemicals and metals [54,55], and may not specifically release heavy metals to the surrounding soil.

### 4.2. Impacts of Poverty on Soil Contamination

Our analyses showed inconsistent and generally insignificant effects of poverty on the heavy metal contamination of soil. We observed a significant positive relationship between poverty and cadmium contamination, which agreed with many past studies that reported elevated soil–metal contamination for poverty filled areas [2,21,25,27]. In further analysis, we found that poverty-stricken individuals were more likely to be directly exposed to chromium, selenium, and silver (Appendix A), but indirectly exposed to other metals via closer proximity to emission sources. Hence, the finding that poverty has both direct and indirect effects on select heavy metal contamination in soil, but does not display any significant total effects (Table 4). Specifically, our results showed that there was a significant negative direct effect for selenium. We also see a significant positive indirect effect for selenium. One reason for seeing both a significant negative direct and positive indirect effect is due to the interference of the built environment characteristics, which play a role as mediators. As selenium has a wide distribution in almost all parent materials in the environment [56]; it can easily be leached into the soil. Hence, the positive indirect effect. In addition, plants are highly effective in removing selenium from contaminated sites, due to scavenging and their copious root systems [56]. This is another reason we see a negative direct effect.

### 4.3. Disproportionate Exposure of Minority to Soil Heavy Metals

Regarding minority, modeling results showed significant increased risk of exposure to heavy metal contamination for arsenic, cadmium, and lead 13.5%, 5.5%, 10.6%; (*p*-values < 0.01)]. These findings agree with past studies that minorities have increased exposure to soil–metal contamination [2,25,26]. Additional analysis determined that minority individuals are directly affected by arsenic and lead. Furthermore, depending on minority proximity to emissions, there is also increased exposure of barium (Table 4). It is expected that we see significant association through proximity to emissions sources in the environment. As mentioned earlier, proximity to emission sources is a major cause of soil contamination. An interesting result is the direct effects; race alone can increase exposure to select heavy metal contamination. A few explanations of the fact that we see significant direct effects for arsenic and lead could be due to presence of heavy metals in everyday life via the use of cosmetics, ingestion of fish or vegetables, use of pesticides, fertilizers, or detergents, pesticide use, contaminated irrigate water, ethnic foods, candy wrappers, certain spices, etc. [25,57,58,59]. In addition, recent research has shown that products (e.g., hair care, beauty, personal care, etc.) made specifically for minority groups are more hazardous and can contain harmful pollutants [60,61,62]. These everyday products can leach into waste water and soil, contaminating the environment with pollutants such as heavy metals [63,64].

### 4.4. Implications of Environmental Racism in the U.S.

Through our findings we have provided evidence of racial injustice in environmental pollution of heavy metals in soil. As social scientists and epidemiologists have long understood that racism is a fundamental cause of disease that operates through complex, ever-changing mechanisms [65,66], there is a need to address the unequal distribution of pollution in urban areas. Our findings can be shared with policy makers informing them of this serious environmental justice issue, to help create interventions and movements that can engage in aggressive primary prevention efforts that go further than soil remediation and urban gardening. New mechanisms are needed to decrease disparity in environmental exposures. Our findings can provide guidance for future interventions for vulnerable populations, such as an antiracist agenda with a special focus on soil contamination regulations that can be implemented across multiple scales, local, regional, and national, to decrease the occurrence of environmental racism in the U.S.

Our study also alludes to the notion that race and poverty are not interchangeable. Past studies that have researched disparities in soil–metal contamination, have not differentiated between race and income [2,26]. Thus, we are able to bring new information to the forefront regarding disparities in soil–metal contamination. Our results showed significant racial disparities but generally insignificant economic disparities in soil heavy metal exposures. This finding agrees with past research that has determined, though minorities are overrepresented among lower SES groups, race and SES have independent effects on health outcomes [67,68]. Furthermore, many studies have found that significant health inequalities between black and white Americans remain when SES is controlled [65,69,70]. Our results infer that race, more so than SES, plays a major role in exposure to heavy metal contamination, alluding to a significant racial disparity EJ problem. Hence, we can conclude the possibility of environmental racism, a term used to describe racial discrimination in environmental policy-making and the enforcement of regulations and laws, the deliberate targeting of communities of color for toxic waste facilities, the official sanctioning of the presence of life-threatening poisons and pollutants for communities of color, and the history of excluding people of color from leadership of the environmental movement [71,72].

### 4.5. Limitations

We have recognized several limitations, mainly from the original data collection. The small inner-city areas did not fully present the spatial variability or all the populations. Industrial facilities used to cluster in inner cities, which may have caused elevated soil contamination [21,25]. The urban sprawl has displaced middle- and upper-class people to the suburbs, leaving low-income and minority people in central cities [21,73]. The original soil sampling missed the suburban areas that typically have more mid- and high-income populations and fewer emission sources. The study areas explain the insignificant disparities within each city (Appendix A). The soil sampling was a one-time sampling without collecting repeated samples. Although heavy metals are known to be stable in soil [74,75,76], the snapshot samples may not represent long-term exposures among local populations. The laboratory analysis only measured total metal concentrations without quantifying the assimilable fractions, the part available for body absorption and determining the toxicity. In terms of the disparity analysis, we only considered proximity to emission sources, which were presumably related to soil contamination. No information was available in the EJScreen database about whether the sources were specific to heavy metals. Moreover, studies have found other built features that cause soil–metal contamination, e.g., vacant properties [77], old housing [24,78], interior and exterior paints [79], and refuse incinerators [80].

## 5. Conclusions

This study confirmed racial disparities in the exposure to soil heavy metals. Higher heavy metal concentrations are detected in the soil of minority-concentrated neighborhoods in the Southeastern U.S. This is the first study to assess EJ and soil contamination across this region. This study is also unique in that it explored the mediating effects of the built environment factors in the context of environmental disparity. and differentiated between race and SES in the analysis. More research is needed to determine if this is a regional EJ issue or an EJ issue across the country. To better understand direct exposure of heavy metals in everyday life, future research could also use biomarkers to assess heavy metal exposure in individuals. Researching this EJ issue can lead to not only an improvement in population health, but also a reduction in racial inequalities across the U.S.

## Figures and Tables

**Figure 1 ijerph-19-01105-f001:**
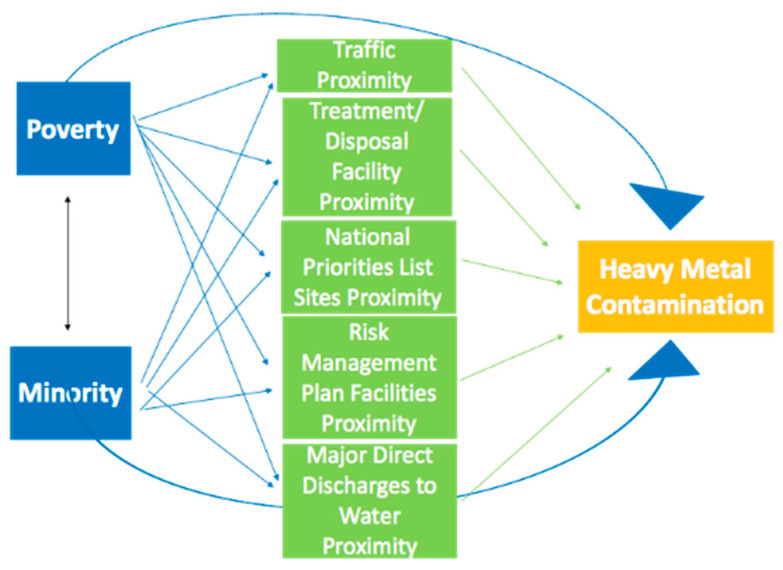
Path diagram of the relationship among income, race, built environment, and soil heavy metal contamination. Figure shows the hypothesized paths to examine the comparative strength of poverty and minority and environmental factors (proximity to traffic, treatment storage and disposal facilities, major and direct discharges to water, national priority sites, and risk management plan facilities) on exposure to heavy metals. Notes: Double-headed arrows indicate covariance. Single-headed arrows indicate path coefficient in relation to heavy metal contamination.

**Figure 2 ijerph-19-01105-f002:**
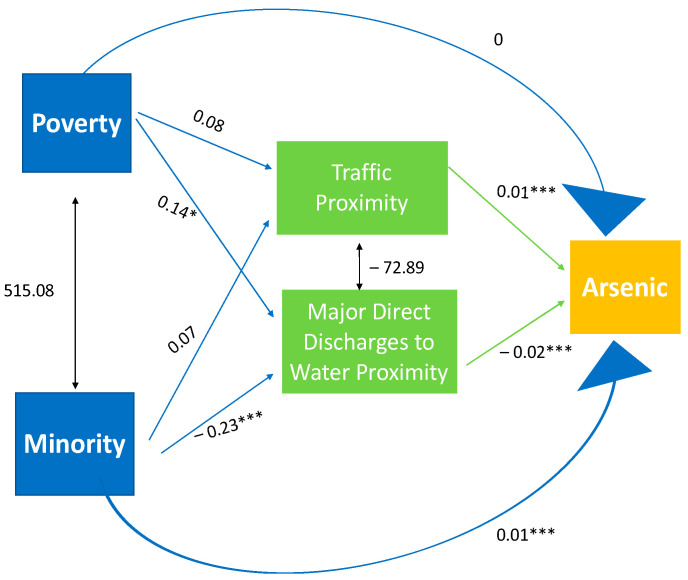
Standardized path coefficients for trimmed path diagram of heavy metal contamination. The figure shows the trimmed path model with nonsignificant paths deleted from model. This depiction clearly displays which paths, poverty, minority, and environmental factors are related to increased exposure to arsenic. Notes: *** indicate significance at <0.001; * indicate significance at <0.05. Double-headed arrows indicate covariance. Single-headed arrows indicate path coefficient in relation to arsenic contamination.

**Table 1 ijerph-19-01105-t001:** Descriptive statistics of minority and poverty in (n = 423) in the Southern U.S.

		Poverty			Minority	
City	Mean	Median	Mode	Mean	Median	Mode
Chattanooga	68.74	80.00	34.00	69.74	75.00	33.00
Columbia	71.17	75.00	53.00	66.06	71.00	76.00
Gainesville	65.36	84.00	87.00	53.60	51.00	72.00
Lexington	46.91	41.00	24.00	68.47	75.00	75.00
Louisville	68.42	77.00	68.00	78.91	91.00	96.00
Memphis	86.87	92.00	98.00	87.62	96.00	97.00
Raleigh	49.38	42.00	65.00	59.32	69.00	1.00
Winston-Salem	64.96	67.00	99.00	64.13	79.00	87.00

**Table 2 ijerph-19-01105-t002:** Concentrations of heavy metal levels [mg/kg] in soil (n = 423) in the Southern U.S.

Heavy Metal	Mean	Median	Max	Min	Std. Dev.	COV (%)
Arsenic	5.14	2.40	270.00	0.20	14.96	291
Barium	86.76	70.50	890.00	1.40	81.83	94
Cadmium	0.30	0.20	3.80	0.09	0.35	117
Chromium	13.83	12.00	63.00	1.10	9.32	67
Lead	95.82	44.50	1400.00	1.70	154.17	161
Selenium	0.68	0.40	2.60	0.36	0.69	101
Silver	0.19	0.10	2.20	0.09	0.24	126

**Table 3 ijerph-19-01105-t003:** Unadjusted association between soil–metal contamination and rankings of poverty and minority in the Southern U.S.

Metal	Ranking of Poverty (n = 423)			Ranking of Minority (n = 423)		
	Estimate ^1^	%Change ^2^	*p*-Value	Estimate ^1^	%Change ^2^	*p*-Value
Arsenic	1.047	4.7%	0.037	1.134	13.5%	<0.0001 *
Barium	−1.006	−0.6%	0.722	1.036	3.6%	0.048
Cadmium	1.047	4.7%	0.0002 *^,3^	1.055	5.5%	<0.0001 *
Chromium	−1.008	−0.8%	0.516	1.018	1.8%	0.149
Lead	1.054	5.4%	0.011	1.106	10.6%	<0.0001 *
Selenium	1.002	0.2%	0.830	−1.003	−0.3%	0.773
Silver	1.013	1.3%	0.254	1.009	0.9%	0.463

Notes: ^1.^ Estimate is β1 from the crude model: log [Metal Concentration] = β0 + β1 × Ranking of Poverty (or Minority). β1 > 0 suggests a positive association, and vice versa. ^2.^ Percent change means percent change in the metal concentration per 10-percentile change in the ranking of poverty or minority. ^3.^ * indicate significance at *p*-value < 0.01.

**Table 4 ijerph-19-01105-t004:** Direct and indirect effects of major exposure variables in a hypothesized model.

Metal	Ranking of Poverty (n = 423)			Ranking of Minority (n = 423)		
	Direct	Indirect	Total	Direct	Indirect	Total
Arsenic	−0.14 *	0.04	−0.10	0.14 *	0.18 *	0.32 *
Barium	−0.11	−0.02	−0.13	0.06	0.12 *	0.18 *
Cadmium	−0.06	0.15 *	0.09	0.08	0.07	0.15
Chromium	−0.11	−0.01	−0.12	0.11	0.04	0.15
Lead	−0.05	0.03	−0.02	0.15 *	0.09 *	0.24 *
Selenium	−0.20 *	0.23 *	0.03	−0.03	−0.004	−0.03
Silver	−0.13	0.18 *	0.05	−0.02	0.02	0.002

Notes: * indicate significance at *p*-value < 0.01.

## Data Availability

Data available in a publicly accessible repository that does not issue DOIs. Publicly available datasets were analyzed in this study. This data can be found here: [https://www.epa.gov/ejscreen/download-ejscreen-data; https://www.epa.gov/risk/regional-urban-background-study] (accessed on: 1 December 2021).

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
