# Peer review of "Racial Disparities in the Heavy Metal Contamination of Urban Soil in the Southeastern United States"

_ijerph, 2022, doi:10.3390/ijerph19031105_

Round 1

Reviewer 1 Report

This manuscript presented a study exploring potential disparities in heavy metal contamination of urban soil among communities with different socioeconomic status (SES) and racial composition in the Southeastern United States. The results confirmed racial disparities in urban soil heavy metal contamination in the study area, but less so for SES. Another unique point of this study is that it explored the mediating effects of the built environment factors in the context of environmental disparity and differentiated between race and SES in analysis. Given that similar studies are scarce, the information provided by this study are particularly valuable for both researchers and policy makers. The reviewer would like to recommend publication of this manuscript after some writing corrections, as detailed below:

  1. Line 72, abbreviation as SES has already been defined in earlier text.
  2. Figure S1 is a quite rough, please modify it to the format of Figure 2 so readers can follow better.
  3. Line 256-257, grammatically wrong.

Author Response

We would like to thank the reviewer for their interest in our manuscript and the insightful, constructive, and careful suggestions and comments that improved our manuscript.

Reviewer 2 Report

Title of manuscript – to accept

Abstract – correct

Keywords – correct

Introduction

Please give heavy metal definition

Do Barium and Selenium are heavy metals ?

You can add map on residents, inhibitants plus contaminations of environment ?

Material and methods

Correct, because it determined by EPA,

Please add information about CRM cert. Ref. Mater....

Results

Correct

Discussion

You can mention about remediation of contamination grounds from studies areas, maybe revil

Conclusion

Correct , please remove references

Author Response

We thank the reviewer for the positive comments on these sections.

Reviewer 3 Report

The paper seems good to me, with a very interesting theme, but I am going to make some comments or suggestions that I think could improve the paper:

The title includes “Racial disparities… ..” and no information is given on minorities and racial groups. It would be interesting to provide a summary of the demographic, socioeconomic and racial data and not only give the reference EPA's, 2016 EJScreen data set [33]. It would be highly recommended to give a summary to the reader.

The bibliography is perhaps too large and some less relevant works could be eliminated.

It is not specified in the methodology if an acid digestion is carried out to determine the content of heavy metals. Is aqua regia used?

It was not sampled in city centers, only in the outskirts, perhaps it would have been interesting to know the level of pollution in city centers as well, which is also where the majority of the  population lives.

Rather than the total values ​​of heavy metals, it would be better to know their assimilable or available value. It would be necessary to know the pH of the soils that influences this availability. The total values ​​are not the best indicator of contamination since it is the available fraction is the one that is toxic and dangerous for health.

Figures 1 and 2 are almost illegible, they are very small.

Lines 204 to 242 drag a smaller font size, from the bottom of Figure 2.

Author Response

We thank the reviewer for the positive comments. We appreciate the valuable comments and suggestions, which are very helpful for improving the quality of this manuscript. We have carefully reviewed all the questions and suggestions and made corresponding corrections or changes.

Round 2

Reviewer 3 Report

The authors have listened to all my suggestions and have substantially improved the paper to be able to publish it.